# Phenotypic Spectrum and Molecular Findings in 17 ATR-X Syndrome Italian Patients: Some New Insights

**DOI:** 10.3390/genes13101792

**Published:** 2022-10-04

**Authors:** Alessandro Vaisfeld, Sara Taormina, Alessandro Simonati, Giovanni Neri

**Affiliations:** 1Department of Medical and Surgical Sciences (DIMEC), University of Bologna, 40138 Bologna, Italy; 2Institute of Genomic Medicine, Catholic University School of Medicine, 00168 Rome, Italy; 3Unit of Medical Genetics, Azienda Ospedaliero-Universitaria di Bologna, IRCCS, 40138 Bologna, Italy; 4Department of Surgery, Dentistry, Paediatrics and Gynaecology, University of Verona, 37129 Verona, Italy

**Keywords:** ATR-X syndrome, ATRX, XLID, chromatin remodelers, neurodevelopment disorders

## Abstract

ATR-X syndrome is a rare X-linked congenital disorder caused by hypomorphic mutations in the *ATRX* gene. A typical phenotype is well defined, with cognitive impairment, characteristic facial dysmorphism, hypotonia, gastrointestinal, skeletal, urogenital, and hematological anomalies as characteristic features. With a few notable exceptions, general phenotypic differences related to specific ATRX protein domains are not well established and should not be used, at least at the present time, for prognostic purposes. The phenotypic spectrum and genotypic correlations are gradually broadening, mainly due to rapidly increasing accessibility to NGS. In this scenario, it is important to continue describing new patients, illustrating the mode and age of onset of the typical and non-typical features, the classical ones and those tentatively added more recently. This report of well-characterized and mostly unreported patients expands the ATR-X clinical spectrum and emphasizes the importance of better clinical delineation of the condition. We compare our findings to those of the largest ATR-X series reported so far, discussing possible explanations for the different drawn conclusions.

## 1. Introduction

ATR-X syndrome (also known as α-thalassemia X-linked intellectual disability syndrome) is a rare X-linked condition caused by hypomorphic mutations in the *ATRX* gene [1], whose phenotype is typically manifested in males. Most cases are inherited from healthy mothers who are carriers, the majority of whom are reported as having a skewed X-inactivation. In rare cases of mildly affected females, the pattern of X-chromosome inactivation is reported as non-skewed [2]. To our knowledge, only one female with typical ATR-X features and an *ATRX* pathogenic variant has been reported, with a marked skewed X-inactivation pattern and preferential inactivation of the “normal” X [3].

The ATR-X acronym stands for X-linked inheritance, alfa-thalassemia (mild) and cognitive impairment (formerly retardation), diagnostic hallmarks identified in early cases [4,5]. Nowadays, clinical characterization is much more detailed, and the phenotypic spectrum has expanded to include characteristic dysmorphisms, hypotonia, gastrointestinal, skeletal, and urogenital anomalies, variably combined in affected individuals. Moreover, atypical phenotypes have been recently reported, mainly due to the increasing recourse to multigenic analysis [6,7].

The ATRX protein is a master epigenetic regulator involved in chromatin remodeling [8,9,10]. Similar to other ATP-dependent chromatin remodelers, it contains a helicase/ATPase domain located in the C-terminal region together with the MeCP2 binding domain and PML targeting motif.

The death domain-associated protein (DAXX) binding motif, critical for its role in telomeric and pericentromeric heterochromatin organization, is located in a more central region of the protein. The C-terminal and central motifs are missing in the ATRXt isoform, which retains intron 11 by alternative splicing, generating a premature stop codon [11]. The full-length ATRX and ATRXt isoforms both contain the HP1a and the EZH2 binding motifs, as well as the N-terminal ATRX-DNMT3-DNMT3L (ADD) domain, which comprises a GATA-like and a plant homeodomain (PHD) [9,12]. The great majority of the ATR-X syndrome-causing mutations are located within the ADD and the helicase domains [13,14].

Different *ATRX* mutations are also found in many types of pediatric and adult cancers. Unlike the germline variants underlying the syndrome, somatic mutations are distributed along the gene without preferential localization [6,9,15] and are considered to be loss-of-function. They are presumptuously more severe and, therefore, incompatible with embryo development.

Genotype-phenotype correlations have been proposed since the first reports of ATR-X syndrome. For instance, it has been suggested that a more severe psychomotor impairment is caused by variants affecting the PHD-like region of the ADD domain and that severe urogenital abnormalities are more frequent in cases where mutations are close to the C-terminus [13,14]. However, specific phenotypic differences related to specific ATRX protein domains are not well established and should not be used at this moment for prognostic purposes [6]. On the other hand, a significant correlation has been established between milder phenotypes and premature stop codons, such as the p.R37*, where translation is initiated from a different methionine residue and a smaller and less abundant protein is produced [16,17]. Likewise, a higher risk of osteosarcoma has been reported for carriers of mutations in the C-terminal domain [18,19,20]. These associations may be taken into consideration for prognostic purposes, even though the limited number of known cases does not allow for definitive conclusions.

For this reason, it is important to continue describing new patients. Additionally, given the lack of substantial data in non-pediatric cohorts, reporting the natural history of the syndrome in adults will unravel the mode and age of onset of the typical and non-typical features, the classical ones and those added more recently.

## 2. Materials and Methods

We describe clinical and molecular characteristics of 17 Italian patients with ATR-X syndrome, afferent to the Italian ATRX Family support group. Of these, 15 were previously unreported. We decided to include the two brothers reported by Gibbons et al. in 2008, still in strict follow-up, since nowadays they are among the oldest reported subjects. Participating patients are not described singularly, but rather as a cohort. Where feasible, the collected clinical and molecular information was reported and discussed as aggregate data.

The only inclusion criterion was to have a molecularly confirmed diagnosis of ATR-X syndrome. All patients were included in the study without an age limit after the release of informed consent by parents or guardians.

Some patients received the diagnosis directly from us, while others came to our attention through the family support group with an already known *ATRX* mutation. The original intent was to personally evaluate all individuals. However, this was not possible for some of the patients because of the restrictions caused by the COVID-19 pandemic. For these cases, we proceeded to contact the families by telephone to collect the necessary information through an ad hoc created data collection form. We asked the caregivers to complete those forms with the support of the patient’s referring physician, as well as provide photographic and other clinical documentation of the patient. In each case, the collected records were evaluated and verified by us.

Given the study design, we included a patient who died from pneumonia in 2016 at the age of 8 years, with exhaustive clinical records available, including the molecular diagnosis. For this patient, the reported age was that of the last recorded clinical evaluation. Conversely, we did not include three maternal uncles of our patients with a suggestive clinical history who died before molecular analysis was available.

Data of selected patients were entered anonymously into a database, assigning to each patient a sequential number (data available on request).

For the calculation of percentiles related to neonatal measurements, we used the program Inescharts [21]. For the 0–2 years age group, we used the specific WHO growth charts [22], and for the age group above 2 years, we used Cacciari’s charts [23]. Microcephaly and short stature have been considered when an individual’s head circumference and height are below the 3rd percentile for their age and population group.

To describe the reported variants, we used the reference sequence NM_000489.6. The HGMD and LOVD reference databases were used for allelic frequencies and to verify previously reported variants [24,25].

Dysmorphic evaluation of patients was performed by GN and AV based on photographs.

## 3. Results

### 3.1. Molecular Data

A total of 10 different germline mutations were present in this series of 17 ATR-X patients belonging to 14 families (Table 1). A single variant, c.736C > T, reoccurred in six individuals belonging to five different families, thus accounting for over 1/3 of the cases. This was not surprising; this mutation is already known as the most common among ATR-X patients [13]. To the best of our knowledge, seven of the other variants have already been described [8,13,14,26,27,28]. Of these, two different missense variants, the c.6253C > T and the c.6254G > A, involve the same amino acid residue, resulting in different substitutions (Table 1), as part of recurrent missense mutations of the ATRX residue 2085 [14,29]. The remaining two are reported here for the first time. These are a terminal single nucleotide deletion (c.7376del) leading to a premature stop codon and a missense variant (c.658T > A) lying into a codon where two different missense variants have been previously reported [30,31].

Considering only independent cases, the molecular diagnosis was achieved through targeted ATRX analysis in 10/14 of the cases (71%) and through NGS (either WES or multi-gene panels) in the remaining four. However, this data cannot be read without considering the time when the analysis was carried out due to rapidly increasing accessibility to NGS. If we only consider patients born after 2015, only 1/5 of the molecular diagnoses are obtained by targeted analysis.

The age at diagnosis ranges from 1 to 23 years, with a median of approximately 4.5 years. If we do not consider patients born before the *ATRX* gene was identified, the age at diagnosis ranges from 1 to 5 years, with a median of 2.25 years.

Missense variants are the prevailing cause of the syndrome in our cohort, clustered in the ADD region and helicase domain, accounting for approximately 50% and 30% of cases, respectively (Appendix A). Most variants were inherited from carrier mothers, with a ratio of apparently de novo mutations of 2/14 (14%). Extended molecular details are listed in Table 1.

### 3.2. Clinical Features

Table 2 summarizes the main clinical data of the 17 ATR-X patients, ranging in age from 3 to 38 years old, with an average of approximately 15 years. Features included in each category are specified and further detailed in the more comprehensive Table 3. Since intellectual disability and delayed motor milestones are shared by all patients, these features are not shown in the tables, similarly to those ATR-X-associated features not found in any of these patients (e.g., osteosarcomas).

By referring to Figure 1, the facial aspects of patients 2, 3, 4, 5, 6, 7, 8, 10, and 15 have been subjectively defined by us (GN and AV) as highly suggestive of the diagnosis. In these patients, mostly carriers of mutations affecting the ADD domain, the percentage of molecular diagnoses obtained by the targeted analysis was 5/8 (one patient is not counted because they underwent analysis of the already known familial mutation), very similar to the overall percentage (62.5 vs. 71%).

The severity of neurodevelopmental delays and of genital anomalies is the main subject of debate when discussing genotype-phenotype correlations. In this series of patients, the only two cases with a moderate to mild degree of intellectual disability are those carrying mutations on residue 2085. These two patients have acquired, and so far, maintained autonomous walking. Neither presented gastrointestinal issues or severe urogenital anomalies, limited to cryptorchidism in one case.

With regard to urogenital abnormalities, of two brothers carrying a C-terminal mutation (c.7376delT), distal to the helicase domain, one had isolated cryptorchidism and the other had isolated renal agenesis.

Sporadic seizures occurred frequently, clustered in a relatively short period of time and not requiring chronic therapy; much rarer are epileptic diseases with peculiar EEG patterns and drug resistance, reported in this cohort in only two cases: one patient with epileptic spasms and another one with early onset tonic-clonic crises, resistant to anti-epileptic drugs.

Rarely reported skeletal features are present in only two cases: occipital dysraphism in one and a hemivertebra in another. In order to have a comparison with the literature data, clinical categories have been designed in accordance with those of the largest ATR-X patient series [6,14].

## 4. Discussion

Considering both the type and position of mutations within the *ATRX* gene, this new series of patients is consistent with the literature data: missense mutations, generally located in the ADD or in the SNF2-like/helicase domains, are the most prevalent. However, this prevalence may change in the future, as a growing number of non-typical cases will be diagnosed by NGS.

In fact, the trend observed in recent years is clearly in the direction of using NGS instead of targeted analysis even when the phenotype is recognizable. Expectedly, the diagnostic yield of mild or less typical cases will increase, redefining the phenotypic and genotypic spectrum of the ATR-X syndrome.

In the context of a fairly composite clinical picture, intellectual disability emerges as a constant feature, and none of the other manifestations requires such a care load for the neuropsychic profile. However, severe behavioral issues that require drugs are rare (at least in this cohort), possibly facilitated by environmental factors.

Our results indicate that certain anomalies with an impact on long-term prognoses, such as gastrointestinal issues and long-lasting hypotonia, are more prevalent in this series than in other series [6]. Among the less frequently reported features, osteosarcoma occurrence should be carefully evaluated. Given that the median age of osteosarcoma appearance for ATR-X patients is not known, it is important to specify that most patients in this case study were under 20 years old at the time of examination, 75% of the total if we only consider patients with variants in the C-terminal domain, thought to correlate with osteosarcoma predisposition. Therefore, we cannot rule out the possibility that cases of osteosarcoma will eventually appear in our cohort.

Similarly, we should pay attention to age differences when reporting endophenotypes, such as epilepsy and osteoporosis, whose appearance may be age-dependent. For instance, in our case, 3/17 of the patients had early osteoporosis, all of them belonging to the older age group, a proportion that is likely to grow as the average age of the cohort increases.

Facial phenotype is on a continuum and is always detectable to some degree, but only in 53% of cases is it highly suggestive of the condition, at least according to our judgement (Table 3). We are aware that this evaluation is somehow personal/subjective. Nevertheless, we observed a certain prevalence of typical traits in carriers of mutations in the ADD domain.

As to prenatal records, in addition to the prevalence of decreased fetal movements, we underline the high frequency of preterm birth, occurring in approximately 1/3 of cases, which differs considerably from the 7% observed in the general Italian population.

Regarding genotype-phenotype correlations, the numbers are clearly too small for any firm conclusion to be drawn. Nonetheless, we note that no significant differences emerged in terms of severity of psychomotor impairment and/or of urogenital abnormalities when considering the involvement of different domains in the mutant protein. What needs to be reported is an apparent correlation between a milder phenotype and mutations involving residue 2085, which is included in the helicase domain. In accordance with what was reported for carriers of such mutations [29], our two patients acquired better motor skills than normally observed in ATR-X individuals, had less pronounced facial traits, and the absence of severe urogenital anomalies. In addition, no gastrointestinal or seizure issues were reported to date, at the ages of 14 and 7 years, respectively (individual features are available on request).

One can reasonably speculate that the contrasting genotype-phenotype correlations reached in different series of patients [6,13,14] may be partly due to the different prevalence of the 2085 residue changes in this series.

In conclusion, the ATR-X phenotypic spectrum and genotype-phenotype correlations are far away from being fully elucidated. The increasing use of NGS in patients with essentially isolated intellectual disability is gradually showing us the mild extremes of the ATR-X phenotypic spectrum. In this scenario, it is important to continue reporting a well-characterized series of patients.

Obviously, results from large NGS panels in atypical patients must be handled with care, paying particular attention to the interpretation of those variants located outside the highly conserved ADD and helicase domain. The causal role of any observed variants should not be taken for granted.

## Figures and Tables

**Figure 1 genes-13-01792-f001:**
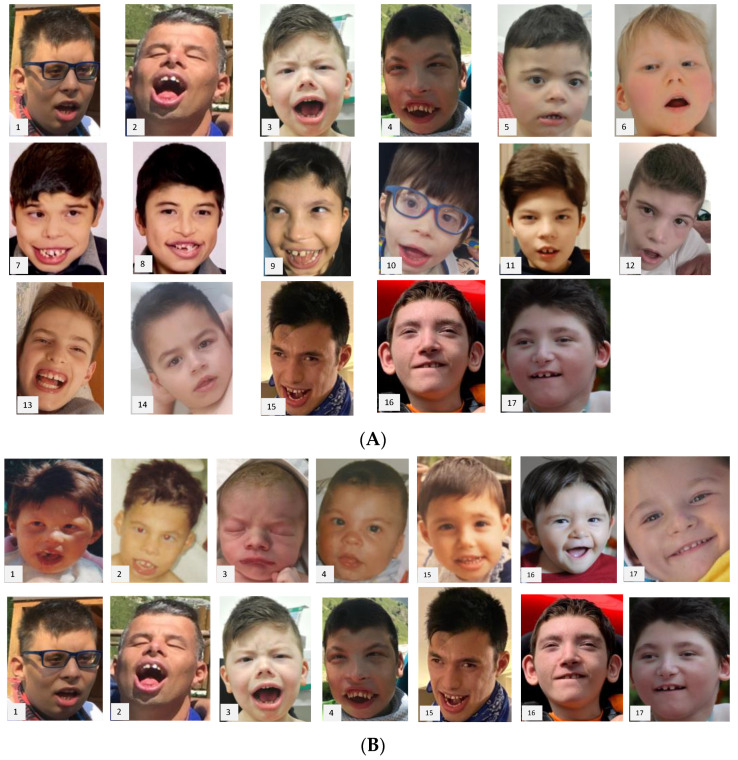
**Facial aspect of patients.** For molecular matches, the number refers to patients ID reported in Table 1. (**A**) Pictures of all patients at current age. (**B**) Where available, pictures of patients at younger (top) and current ages (bottom).

**Table 1 genes-13-01792-t001:** Molecular details of the 10 variants represented in this cohort.

cDNA Variant	Protein Change	Located on Hot-Spot Domains	Variant Type	N. of Carriers (Independent Cases)	Patient ID	Previously Reported in
*c.187G > T*	p.E63*	No	Nonsense	2 (1)	1–2	[13]
*c.536A > G*	p.178_198del ^α^	Yes (ADD)	Missense	1 (1)	3	[13,14,27]
*c.658T > A*	p.C220S	Yes (ADD)	Missense	1 (1)	4	NR
*c.736C > T*	p.R246C	Yes (ADD)	Missense	6 (5)	5–10	[13] ^β^
*c.1727C > A*	p.S576*	No	Nonsense	1 (1)	11	[8,26]
*c.5273-5C > G*	p.Y1758fs	Yes (Helicase)	Splice-site variant	1 (1)	12	[13]
*c.6253C > T*	p.R2085C	Yes (Helicase)	Missense	1 (1)	13	[28,29]
*c.6254G > A*	p.R2085H	Yes (Helicase)	Missense	1 (1)	14	[14,29]
*c.6508A > G*	p.T2170A	Yes (Helicase)	Missense	1 (1)	15	[28]
*c.7376del*	p.M2459Sfs*21	No	Frameshift small del	2 (1)	16–17	NR

NR: not previously reported. ^α:^ functional studies of this variant suggest the creation of a cryptic splice donor site, leading to deletion of 63 nucleotides and 21 (178_198) amino acids deletion. ^β^: mentioned in several reports, only one of which is listed in the Table.

**Table 2 genes-13-01792-t002:** Frequency of total and domain-specific clinical features.

Clinical Features	Total Cases	ADD Domain ^a^	Helicase Domain ^b^
Highly suggestive facial traits	9/17	7/8	1/4
Urogenital anomalies	14/17	7/8	3/4
Skeletal anomalies	15/17	8/8	2/4
Gastrointestinal problems	15/17	8/8	2/4
Hematological anomalies	12/17	6/8	2/4
Heart defects	3/17	3/8	0/4
CNS anomalies	10/17	5/8	3/4

Features are gathered together in large categories, further detailed in Table 3. ^a^: eight mutations in the ADD domain. ^b^: four mutations in the helicase domain.

**Table 3 genes-13-01792-t003:** Detailed clinical features and their frequency in the overall population of patients.

	Tot	freq
**Prenatal and birth**		
Decreased fetal movements	10	59%
Pre-term birth (GA<37 w)	6	35.5%
C-section	8	47%
OFC<5th percentile	7	41%
Length<5th percentile	4	23.5%
**Genitourinary**		
Cryptorchidism	12	70.5%
Small penis	3	17.5%
Hypospadias	2	12%
Shawl scrotum	2	12%
Kidney anomalies ^a^	4	23.5%
**Neurologic**		
Severe intellectual disability	15	88%
Hypotonia	15	88%
Apraxia	6	35.5%
Seizures	9	53%
**Gastrointestinal**		
Dysphagia	12	70.5%
Gastrointestinal reflux	14	82.5%
Gastric pseudo-volvulus	2	12%
Esophagitis/peptic ulcer	2	12%
Colonic hypoganglionosis	4	23.5%
**Skeletal**		
Microcephaly	12	80% *
Short stature	11	64.5%
Scoliosis/Kyphosis	10	59%
Hand/foot anomalies ^b^	11	64.5%
Pes planus/varus/valgus	5	29.5%
**Heart**		
Septal defects	2	12%
Dilated/stenotic aorta	1	6%
**Others**		
Coloboma of iris	1	6%
Other ocular issues ^c^	5	29.5%
Hypoacusia	5	29.5%
Neuroimaging signs ^d^	10	59%
Dysthyroidism	3	17.5%
Obstructive sleep apnoea syndrome	4	23.5%
Osteoporosis	3	17.5%
Umbilical hernia	1	6%

^a^: includes hydronephrosis, pyelectasis, hypoplasia, or agenesis. ^b^: includes clinodactily, camptodactily, brachidactily, and overlapping digits. ^c^: includes severe myopia, astigmatism, strabismus, or cataract. ^d^: includes mild cerebral atrophy, corpus callosum agenesis, white matter anomalies, brainstem, or olfactory bulb hypoplasia. *: head circumference measurement was not available for two individuals; frequency refers a total of 15 patients. Unlisted ATR-X associated features (such as osteosarcomas or asplenia) should be intended as not present in this patient series.

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
