# Peer review of "Phenotypic Spectrum and Molecular Findings in 17 ATR-X Syndrome Italian Patients: Some New Insights"

_genes, 2022, doi:10.3390/genes13101792_

Round 1
Reviewer 1 Report
Phenotypic spectrum and molecular findings in 17 ATR-X syndrome Italian patients: some new insights.
Vaisfeld A. et al
This is nicely written report of 17 cases which add substantially to the number of reported cases, it is in good agreement with previous reports and helps underline the principal features of this condition. In the future it will be interesting to compare the findings here where most cases were identified phenotypically with a report based on cases identified solely by NGS. It is somewhat disappointing that alpha thalassaemia was not looked for in this cohort as this remains a useful functional test especially when the pathology of a variant is uncertain.
Specific points:
Pg 1 line 35: It should be alpha not alfa.
In Table 1 c536A>G actually leads to abnormal splicing: c.532_594del; p.V178_198del. This was originally reported (using different numbering) by Picketts et al 1996 HMG 5:1899 and then by Gibbons et al 1997 Nat Genet 17:146. There have been several other reports where it has mistakenly been reported as a missense mutation.
In Table 3 there is another typo: lenght instead of length.
Author Response
Point 1: Pg 1 line 35: It should be alpha not alfa.
Response to 1: Corrected as suggested
Point 2: In Table 1 c536A>G actually leads to abnormal splicing: c.532_594del; p.V178_198del. This was originally reported (using different numbering) by Picketts et al 1996 HMG 5:1899 and then by Gibbons et al 1997 Nat Genet 17:146. There have been several other reports where it has mistakenly been reported as a missense mutation.
Response 2: We thank the reviewer for this important point. In table 1, we maintain the cDNA nomenclature, while correcting its consequence on the protein (p.178_198del), further specified in the headings.
Point 3: In Table 3 there is another typo: lenght instead of length.
Response 3: Also corrected as suggested

Reviewer 2 Report
This manuscript is a review report of 17 Italian ATR-X syndrome patients. ATR-X syndrome is very rare syndrome, and more than 200 patients are diagnosed all over the world. Clinical reports like this are important to accumulate clinical information.
1. The term “Alpha-Thalassemia X-linked Intellectual Disability Syndrome” should be used with “ATR-X syndrome”, because the term is registered in many databases, including GeneReviews, or XLID genetic Research in Greenwood Genetic Center, as you know.
2. How many years does it take to diagnose these 17 patients as ATR-X patients in Italy? For example, more than 100 patients have been diagnosed in Japan in this quarter century.
3. Are these 17 patients almost all cases diagnosed in Italy, or just part of them? Do you have any system to register ATR-X patients in Italy?
4. Do you have an estimated prevalence of ATR-X syndrome in your country?
5. I agree that the introducing NGS can find many variants in ATRX in atypical cases, and broaden the clinical spectrum of ATRX-related phenotype. In this manuscript, it is not clear about clinical features of atypical cases. How “relatively mild” are the cases with variants on residue 2085.
6. In this report all clinical information of 17 patients is described comprehensively, not individually. It is much better to organize their clinical information in a table individually. We do not know how old they are now, although we know they were diagnosed between 1 and 26-year-old.
7. What is the difference between targeted ATRX analysis era before 2015 and NGS era after 2015?
8. Figure 1; Individual photos are very fascinating and impressive. It seems better to integrate 1A and 1B.
9. Line 91; Covid should be described as “COVID-19 pandemic”.
10. Line 97; One patient passed away at 8 years old. What is the cause of his death? This is important information. Generally clinical information should be described more precisely.
11. Table 2; “Clinical features”; what and how severe are their Urogenital, Skeletal, Gastrointestinal, Hematological, Heart, and CNS anomalies?
We already have some clinical reviews by Prof. Richard Gibbons. The author should describe clinical information individually and precisely, in order to add new insights by this report.
Author Response
Point 1: The term “Alpha-Thalassemia X-linked Intellectual Disability Syndrome” should be used with “ATR-X syndrome”, because the term is registered in many databases, including GeneReviews, or XLID genetic Research in Greenwood Genetic Center, as you know.
Response 1: We accepted the Reviewer suggestion, specifying in the introduction that “ATR-X syndrome” and “Alpha-Thalassemia X-linked Intellectual Disability Syndrome” are both used. We also included “XLID” term in the keywords, instead of “ID”.
Point 2: How many years does it take to diagnose these 17 patients as ATR-X patients in Italy? For example, more than 100 patients have been diagnosed in Japan in this quarter century.
Response 2: In this patients’ series, the molecular diagnosis timeframe is 18 years (2001-2019). Nonetheless, we should consider that they did not represent all the italian ATR-X patients diagnosed during this period (please, see the next point). Therefore, we could not use them to have an idea of ATR-X syndrome incidence in Italy.
Point 3: Are these 17 patients almost all cases diagnosed in Italy, or just part of them? Do you have any system to register ATR-X patients in Italy?
and
Point 4: Do you have an estimated prevalence of ATR-X syndrome in your country?
Response 3 and 4: We merged questions3 and 4, and provide a single answer.
No, these are not all the cases; we know from colleagues that there are patients not enrolled in our series, but we don’t have a precise estimate of how many we are missing.
The “ATR-X Italia” support group is only 2 years old and it does not cover the entire Italian territory, and we do not have an establish system to register ATR-X patients.
Point 5: I agree that the introducing NGS can find many variants in ATRX in atypical cases, and broaden the clinical spectrum of ATRX-related phenotype. In this manuscript, it is not clear about clinical features of atypical cases. How “relatively mild” are the cases with variants on residue 2085.
Response 5: We have reported on page 4 the phenotypic expression of the two carriers of residue 2085 variants, both in terms of ID, gatrointestinal and urogenital anomalies. We only changed “medium to relatively mild” to “moderate to mild”.
Point 6: In this report all clinical information of 17 patients is described comprehensively, not individually. It is much better to organize their clinical information in a table individually. We do not know how old they are now, although we know they were diagnosed between 1 and 26-year-old.
Response 6: This is a point we have deeply discussed, recognizing ita s a critical issue. As specified in “Material and methods” section, we cocluded thai it would be preferable to describe the cases as a cohort, and their features reported as aggregate data (similar to other published series). In any case, all the individual clinical records are available on request. We would be happy to share the comprehensive database (Excel format) with the Reviewer.
Point 7: What is the difference between targeted ATRX analysis era before 2015 and NGS era after 2015?
Response 7: We addressed this issue on page 3 and pointed out that in our cohort the number of cases diagnosed after 2015 are too few to make a meaningful comparison.
Point 8: Figure 1; Individual photos are very fascinating and impressive. It seems better to integrate 1A and 1B.
Response 8: We agree that it is unfortunate not to have all the patient photos at different ages. We tried to create a single figure, but the yield was somehow chaotic and eventually opted for 1A and 1B separation.
Point 9: Line 91; Covid should be described as “COVID-19 pandemic”.
Response 9: Thank you, the term has been changed as suggested.
Point 10: Line 97; One patient passed away at 8 years old. What is the cause of his death? This is important information. Generally clinical information should be described more precisely.
Response 10: He died for pneumonia complications. The cause of death has been reported on the manuscript on page 3, line 98.
Point 11: Table 2; “Clinical features”; what and how severe are their Urogenital, Skeletal, Gastrointestinal, Hematological, Heart, and CNS anomalies?
Response 11: As specified in the section headings, clinical features are reported as categories in table 2, while further details are reported in table 3. As for the distinction between ADD and helicase domain, our numbers are too small for such a distinction to be made.
Point 12: We already have some clinical reviews by Prof. Richard Gibbons. The author should describe clinical information individually and precisely, in order to add new insights by this report.
Response 12: We are aware of the excellent reviews by Gibbons group (see references). We are simply heading a good amount of new data for a rare syndrome as also recognized by the same reviewer 2.
